# Exploration driven by a medial preoptic circuit facilitates fear extinction in mice

Anna Shin[1,2,5], Jia Ryoo[1,2,5], Kwanhoo Shin[1,2], Junesu Lee[1,2], Seohui Bae[3], Dae-Gun Kim[1,2,4], Sae-Geun Park[1] & Daesoo Kim [1,2✉]

Repetitive exposure to fear-associated targets is a typical treatment for patients with panic or post-traumatic stress disorder (PTSD). The success of exposure therapy depends on the active exploration of a fear-eliciting target despite an innate drive to avoid it. Here, we found that a circuit running from CaMKIIα-positive neurons of the medial preoptic area to the ventral periaqueductal gray (MPA-vPAG) facilitates the exploration of a fear-conditioned zone and subsequent fear extinction in mice. Activation or inhibition of this circuit did not induce preference/avoidance of a specific zone. Repeated entries into the fear-conditioned zone, induced by the motivation to chase a head-mounted object due to MPA-vPAG circuit photostimulation, facilitated fear extinction. Our results show how the brain forms extinction memory against avoidance of a fearful target and suggest a circuit-based mechanism of exposure therapy.

[1] Department of Biological Sciences, KAIST, Daejeon 34141, Republic of Korea. [2] Department of Brain and Cognitive Sciences, KAIST, Daejeon 34141, Republic of Korea. [3] KAIST Institute of Artificial Intelligence, KAIST, Daejeon 34141, Republic of Korea. [4] ACTNOVA, Daejeon 34109, Republic of Korea. [5] These authors contributed equally: Anna Shin, Jia Ryoo. ✉email: daesoo@kaist.ac.kr

Extinction of fear memories requires the acquisition of new memories through exposure to the fear-conditioned target without the associated aversive unconditioned stimulus[1–3]. Prolonged exposure (PE) therapy, which is a well-established therapy for patients suffering from fear memory-associated disorders, like post-traumatic stress disorder (PTSD), takes advantage of the mechanism of fear extinction[4–6]. Studies have revealed some relevant brain areas and circuits involved in the extinction of fear memory[7–16].

During PE therapy, patients are forced to interact with the fear target or recall a fear memory even though this process is painful and induces strong avoidance against fear-eliciting targets, leading to clinical concerns regarding side effects such as exacerbation or dropout[17,18]. To minimize avoidance effects, it is essential to consider the patient's level of endurance and control the strength of exposure of the fear-eliciting target or fear memory recall appropriately[19,20]. Unraveling the neural mechanism that induces motivation to explore a fear-eliciting target, which can reduce this avoidance, may facilitate the development of additional efficient exposure therapies and/or drugs.

The medial preoptic area (MPA) is known to regulate approach behaviors and exploratory interactions with a variety of targets[21–25]. Also, the ventral periaqueductal gray (vPAG), which receives a strong input from the MPA[26,27], is well known to be involved in modulating approaching (fight) or avoiding (flight) behaviors[28,29] and hunting behaviors[30,31]. Recent studies have revealed that MPA projections to the vPAG mediate novelty seeking, exploration, hunting behaviors, and approach to a variety of targets[32–34].

Park et al have shown in a previous study that calcium calmodulin kinase 2 alpha (CaMKIIα)-positive neurons in the MPA increase the exploratory drive toward objects and contribute to hunting behavior through their projections to the vPAG. The authors used this finding to develop a brain-computer interface technology, MPA-induced drive-assisted steering (MIDAS), wherein explorative drive was controlled by MIDAS to guide animals along a predicted pathway during chasing of a head-mounted object[32]. Here, we tested whether MPA-vPAG-induced explorative drive can facilitate approach to a fear-eliciting target and thereby contribute to fear extinction.

## Results

### A focal zone-fear paradigm to evaluate avoidance and approach behaviors.
In conventional fear conditioning or extinction, subject animals are tested in a closed chamber where they are unable to approach or escape. To compare avoidance and approach factors, we designed a focal zone-fear paradigm using a chamber consisting of two zones (Fig. 1a): a safe zone ($40 \times 40 \times 45$ cm) and a shock zone with an electrical grid floor ($30 \times 30 \times 45$ cm). A fear-extinction schedule was utilized (Fig. 1b): On day-1 (pre-test), the basal activity of mice was recorded in the focal fear chamber. On day-2 (post-test), subject mice were trapped in the shock zone and given electrical foot shocks (1 mA, 5-second duration; total, 3 times) (Fig. 1c). We compared their behavior in the chamber for 5 min before (pre) and after (post) the foot shocks (Fig. 1d). Fear conditioning decreased time inside, entries into and locomotion inside the shock zone (Fig. 1e–g) while increasing time spent freezing (Fig. 1h). These data indicate that we can measure the approach and avoidance behavior of mice during focal zone-fear conditioning.

### Exploration of the fear-conditioned zone facilitates fear extinction.
Next, we sought to determine the effect of the exploratory behavior induced by the stimulation of the MPA-

vPAG circuit on fear memory extinction. To stimulate the MPA-vPAG circuit, we injected adeno-associated viruses (AAV) harboring the channelrhodopsin-2 gene (ChR2) under the control of the CaMKIIα promoter into the MPA (Fig. 2a). We then inserted an optic fiber over the vPAG, where the axons of ChR2-expressing MPA neurons terminate (Fig. 2b). AAV-CaMKIIα−mCherry virus was injected in separate control mice (Supplementary Fig. 1a). To conduct the focal zone-fear paradigm task, mice underwent daily exploration sessions from day-2 (post) to day-9: mice received light stimulation in the shock zone for 5 min. Twenty-four hours after each exploration session, mice from both groups were subjected to an exploration test in which they were allowed to roam freely throughout the focal fear chamber for 5 min as the extinction session (Fig. 2c).

During the exploration period, the ChR2 group showed more movement in the shock zone than the control group (Supplementary Fig. 1b, c). This indicated that stimulation of the MPA-vPAG circuit significantly increased the exploratory behavior of mice in a fear-conditioned zone. To assess the effect of MPA-vPAG circuit stimulation on fear extinction, we conducted extinction tests for both groups. The results indicated that the ChR2 group moved more in the shock zone than the control group, with a significant difference seen from day-8 to day-10 (Fig. 2d, e). In addition, significant differences in the time spent in the shock zone (Fig. 2f) were seen from day-5. Significant differences in the number of entries into the shock zone (Fig. 2g) were seen from day-7. The percentage of freezing in the ChR2 group significantly decreased compared to controls from day-4 (Fig. 2h). These results suggest that the increased exploration behavior in the fear-conditioned zone involved the extinction of zone dependent fear memory, and that the fear-extinction effect triggered by stimulating the MPA-vPAG circuit was increased by repeated trials.

To verify whether the fear-extinction effect via MPA-vPAG circuit stimulation also works in a safe zone, which is near the fear-conditioning zone, we performed a modified focal zone-fear conditioning task. Mice received photostimulation in the safe zone instead of the shock zone during the exploration period (Fig. 2i). Photostimulation increased movement in the safe zone of ChR2 mice, exhibiting higher exploration than controls (Supplementary Fig. 2a, b). In the extinction test, the ChR2 group did not show significant differences in movement in the shock zone after conditioning (Fig. 2j). Basal activity was significantly different, but the total movement was unchanged (Supplementary Fig. 2c). The percentage time spent freezing (Fig. 2k), time spent in the shock zone (Supplementary Fig. 2d), and number of entries in to the shock zone (Supplementary Fig. 2e) were unchanged. These results suggest that photoactivation of the MPA-vPAG circuit in the safe zone during the exploration period did not induce a fear extinction effect.

### Photoinhibition of the MPA-vPAG circuit does not affect exploration-mediate fear extinction.
To test to whether activity of the MPA-vPAG circuit is necessary for fear extinction driven by exploration, we performed the focal zone-fear conditioning test while photoinhibiting the circuit using a light-gated chloride channel (eNpHR3.0) (Fig. 3a, b). We bilaterally injected AAV-CaMKIIα-eNpHR-eYFP virus into the MPA and bilaterally implanted optic fibers over the vPAG (Fig. 3c). AAV-CaMKIIα-eGFP virus was injected in control mice (Supplementary Fig. 3a). During the exploration period, there were no statistically significant differences in the distance moved between eNpHR mice and control mice (Supplementary Fig. 3b), whereas on day-2 (post) eNpHR mice moved less and on day-9 eNpHR mice moved more compared to control mice (Supplementary Fig. 3c).

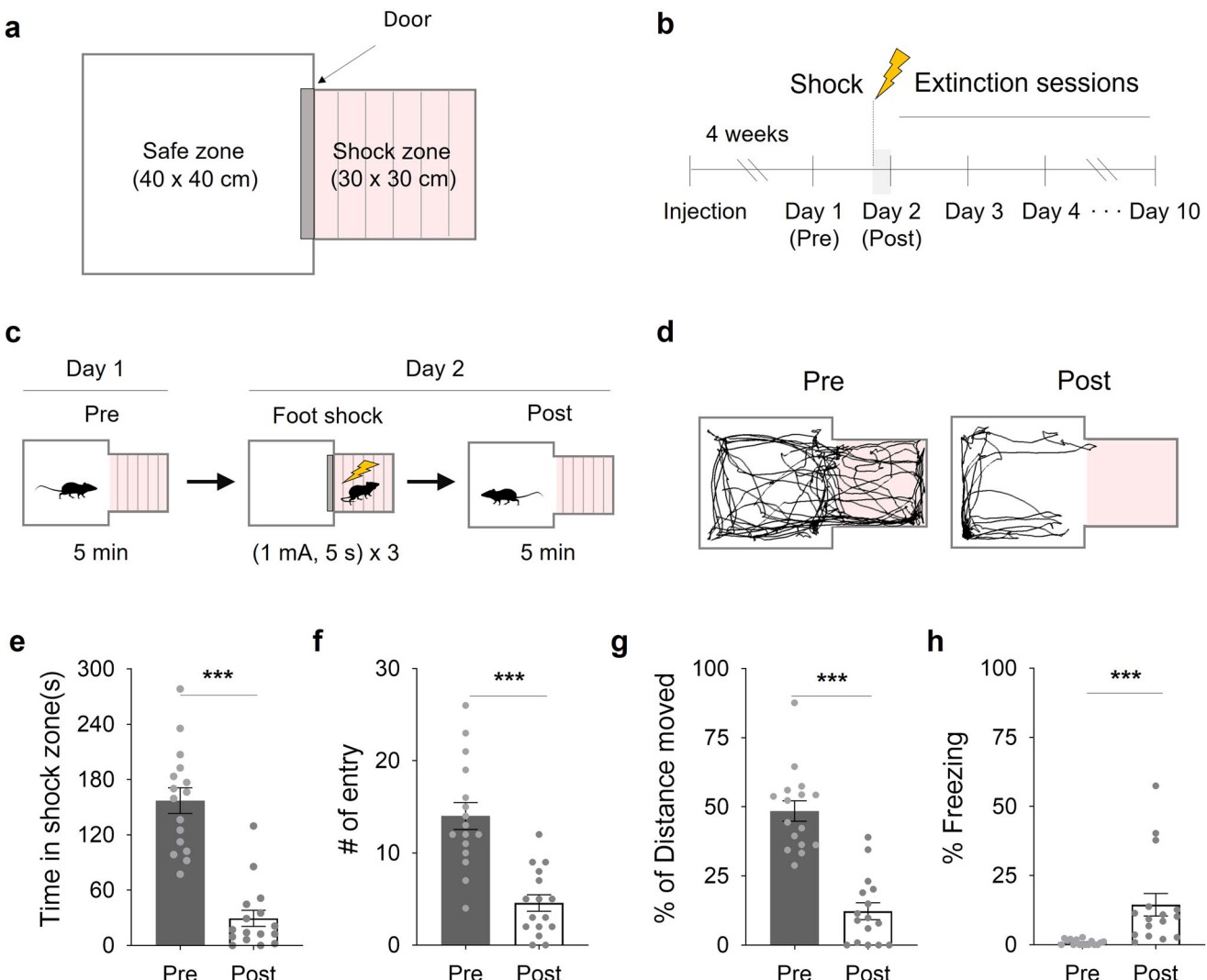

**Fig. 1 A focal zone-fear paradigm for fear conditioning. a** A schematic depiction of the focal zone-fear conditioning. **b** The focal fear-conditioning timeline. **c** The fear-conditioning and pre/post-test schedules. **d** A representative tracking of the center point of a mouse both before (Pre) and after (Post) the shock session. **e** Time in the shock zone before and after the shock session ($n = 16$; paired two-tailed $t$-test,). **f** The number of entries into the shock zone before and after the shock session ($n = 16$; Wilcoxon Signed Rank Test). **g** Freezing percentage before and after the shock session ($n = 16$; Wilcoxon Signed Rank Test). **h** The percentage of distance moved in the shock zone before and after the shock session ($n = 16$; paired two-tailed $t$-test). ***$p < 0.001$. All error bars represent s.e.m. Further information on statistical analyses is given in the supplementary data 1.

Although there was a difference in movement on day-2 (post) and day-9 during the exploration period, no differences in movement (Fig. 3d and Supplementary Fig. 3d), time spent (Fig. 3e), number of entries in the shock zone (Fig. 3f), and percentage time spent freezing (Fig. 3g) were observed between the eNpHR and eGFP groups in the extinction test.

**Photoactivation of the MPA-vPAG circuit facilitates exploration without affecting place preference.** Next, we tested whether the facilitated fear extinction observed in the ChR2 group was due to association of the shock zone with reward, due to photoactivation of the MPA-vPAG circuit. To address this, we examined how photoactivation of the MPA-vPAG circuit affected the results of a conditioned place-preference (CPP) test. In this paradigm, photoactivation was administered in daily training sessions (over 6 days). Mice received photostimulation in zone b but did not receive it in the corridor or in zone a. The effect of CPP was tested on day-8 without any accompanying photostimulation (Fig. 4a, b).

Before photoactivation, we did not observe any differences in the ratio of total time spent in zone b, between ChR2 (MPA-vPAG$^{ChR2}$, CaMKIIα-ChR2-mCherry virus-injected) and control (MPA-vPAG$^{mCherry}$, CaMKIIα-mCherry virus injected) mice (Supplementary Fig. 4a–d). From conditioning day-1 (C1) to conditioning day-6 (C6), the ChR2 group showed preference to zone b while the control group did not (Fig. 4c and Supplementary Fig. 4e–h). However, on day-8, neither group showed any difference in place preference (Fig. 4d–g). These results show that although photoactivation of the MPA-vPAG circuit increases exploration, it does not induce conditioned place preference.

**Photoinhibition of the MPA-vPAG circuit does not affect place preference.** We performed the CPP protocol while photoinhibiting the MPA-vPAG circuit, which was operationalized using mice expressing eNpHR and illumination with an appropriate wavelength of light (Fig. 4h, i). Over the whole series of CPP sessions, MPA-vPAG$^{eNpHR}$ mice showed no significant

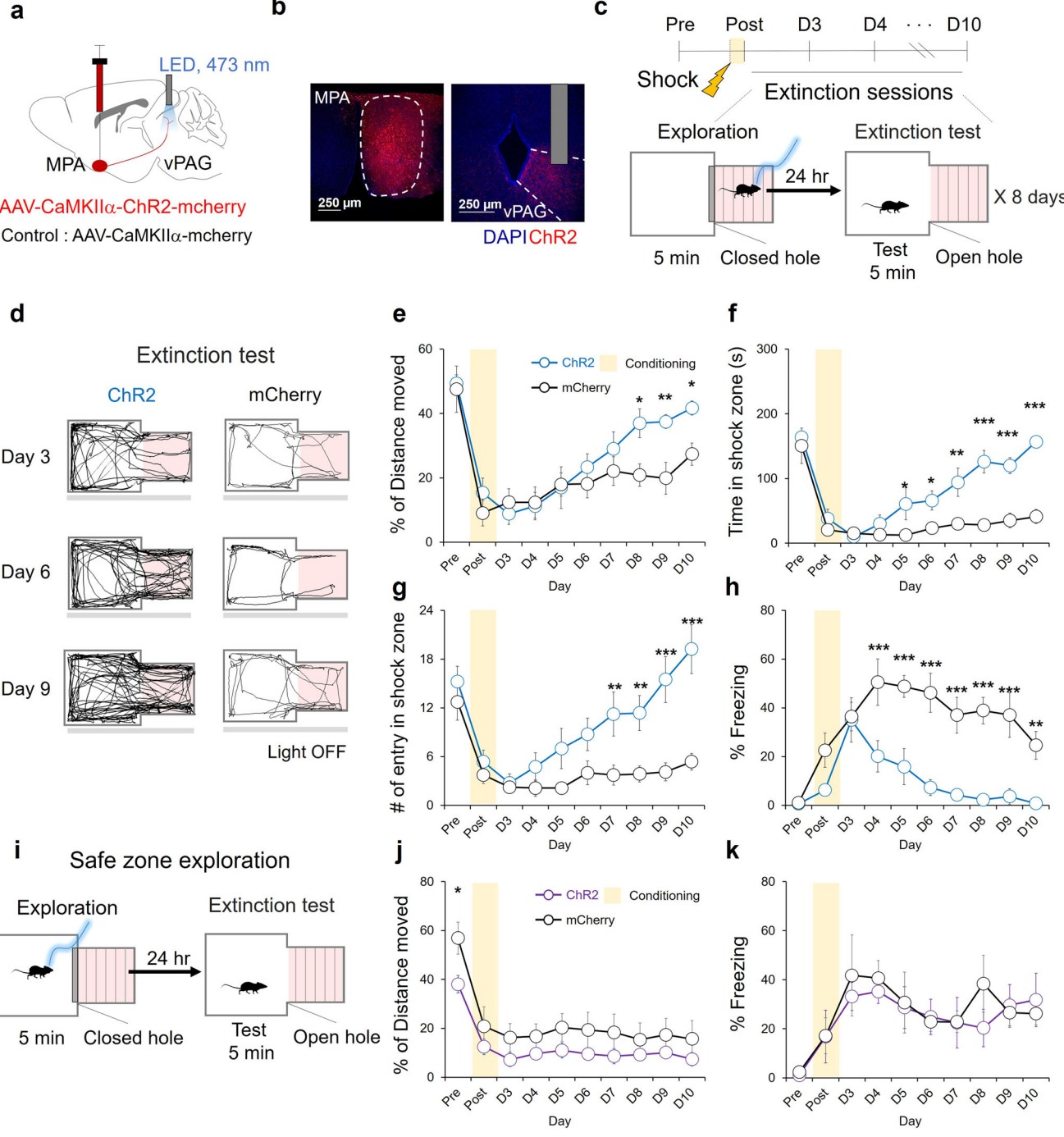

**Fig. 2 Exploration of the fear-conditioned zone induces fear extinction. a** A schematic depiction of virus injection and photostimulation (3 mW, 20 Hz, 5 ms). **b** A representative image of AAV2/9-CaMKIIα-hChR2-mCherry expression (red) in the MPA and the vPAG. Scale bar: 250 μm. **c** Fear extinction protocol with MPA-vPAG circuit stimulation. **d** A representative tracking of the center points of mouse during the extinction test. **e** The percentage of distance moved in the shock zone during the extinction test. The ChR2 group (blue, $n = 8$) and the mCherry group (black, $n = 8$; two-way RM ANOVA test, Holm-Sidak post-hoc analysis). **f** Time in shock zone during extinction test. The ChR2 group (blue, $n = 8$) and the mCherry group (black, $n = 8$; two-way RM ANOVA test, Holm-Sidak post-hoc analysis). **g** The number of entries into the shock-paired zone during the extinction test. The ChR2 group (blue, $n = 8$) and the mCherry group (black, $n = 8$; two-way RM ANOVA test, Holm-Sidak post-hoc analysis). **h** The percentage of freezing during the extinction test. The ChR2 group (blue, $n = 8$) and the mCherry group (black, $n = 8$; two-way RM ANOVA test, Holm-Sidak post-hoc analysis). **i** Modified focal zone-fear conditioning protocol. Photostimulation in a safe zone during exploration. **j** The percentage of distance moved in the shock zone during the extinction test. The ChR2 group (purple, $n = 6$) and the mCherry group (black, $n = 4$; two-way RM ANOVA test, Holm-Sidak post-hoc analysis). **k** The percentage of freezing during the extinction test. The ChR2 group (purple, $n = 6$) and the mCherry group (black, $n = 4$; two-way RM ANOVA test, Holm-Sidak post-hoc analysis). *$p < 0.05$, **$p < 0.01$, ***$p < 0.001$. All error bars represent s.e.m. Further information on statistical analyses is given in the supplementary data 1.

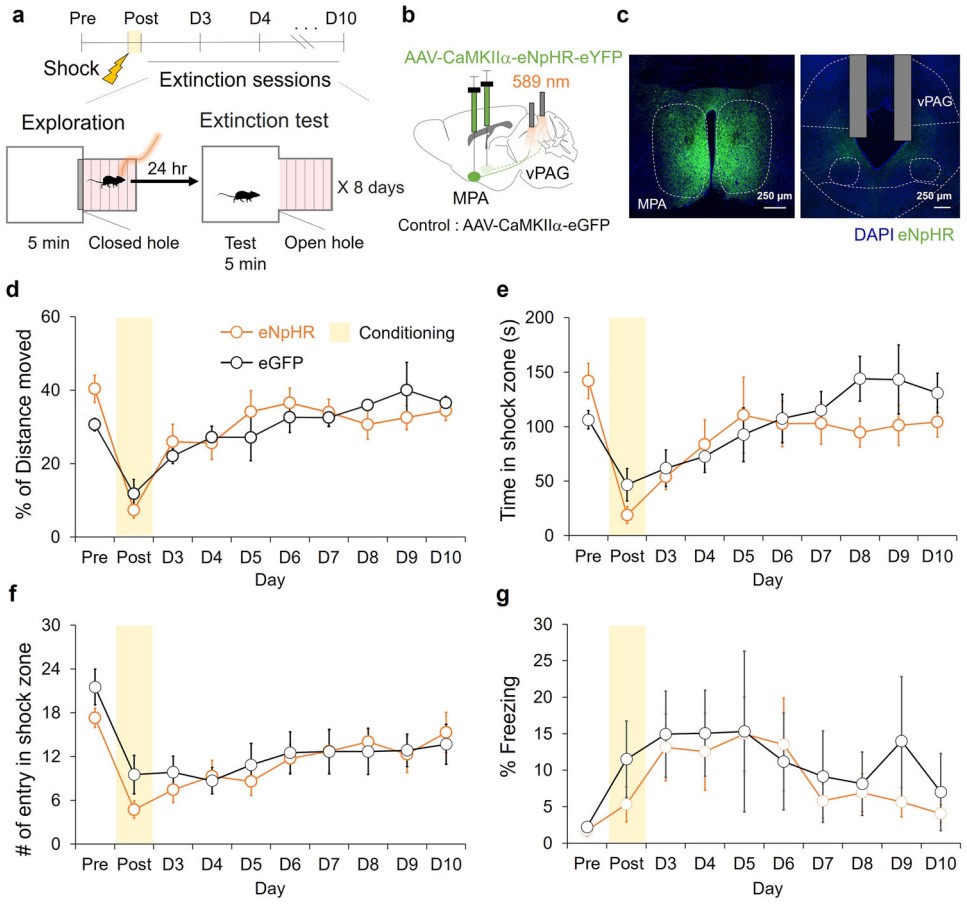

**Fig. 3 Photoinhibition of the MPA-vPAG circuit does not affect exploration mediated fear extinction. a** The focal zone-fear conditioning protocol with MPA-vPAG circuit stimulation. **b** A schematic depiction of virus injection and photostimulation (10 mW, continuous). **c** A representative image of AAV2/5-CaMKIIα-eNpHR3.0-eYFP expression (green) in the MPA and the vPAG. Scale bar: 250 μm. **d** The percentage of distance moved in the shock zone during the extinction test. The eNpHR group (orange, $n = 7$) and the eGFP group (black, $n = 6$; two-way RM ANOVA test, Holm-Sidak post-hoc analysis). **e** Time in shock zone during the extinction test. The eNpHR group (orange, $n = 7$) and the eGFP group (black, $n = 6$; two-way RM ANOVA test, Holm-Sidak post-hoc analysis). **f** The number of entries into the shock-paired zone during the extinction test. The eNpHR group (orange, $n = 7$) and the eGFP group (black, $n = 6$; two-way RM ANOVA test, Holm-Sidak post-hoc analysis). **g** The percentage of freezing during the extinction test. The eNpHR group (orange, $n = 7$) and the eGFP group (black, $n = 6$; two-way RM ANOVA test, Holm-Sidak post-hoc analysis). All error bars represent s.e.m. Further information on statistical analyses is given in the supplementary data 1.

increase in preference or avoidance compared to MPA-vPAG[eGFP] mice (Fig. 4j). Prior to light stimulation, we did not observe any significant increase in the preference or avoidance of a specific chamber for MPA-vPAG[eGFP] mice (CaMKIIα-eGFP virus-injected mice) (Supplementary Fig. 5a, b) or MPA-vPAG[eNpHR] mice (CaMKIIα-eNpHR-eYFP virus-injected mice) (Supplementary Fig. 5c, d). On conditioning day-1, we did not observe any difference in the time spent in zone a vs. zone b under 589 nm illumination for MPA-vPAG[eGFP] mice (Supplementary Fig. 5e, f) or MPA-vPAG[eNpHR] mice (Supplementary Fig. 5g, h). In the test session, neither MPA-vPAG[eGFP] mice (Fig. 4k, l) nor MPA-vPAG[eNpHR] mice (Fig. 4m, n) showed a significant difference in the time spent in zone a vs. zone b. These results indicate that MPA-vPAG circuit inhibition does not affect place preference or aversion.

**Object-guided exploration of the fear-conditioned zone facilitates fear extinction.** We next attempted to induce fear extinction by exposing mice to the shock zone. To increase the number of entries into the fear-conditioned zone, we used MIDAS to guide animal behavior by activating MPA neurons that project to the vPAG to support object-craving behaviors[32]. To stimulate the

MPA-vPAG circuit, we injected AAV-CaMKIIα-ChR2-mCherry into the MPA of mice. We then inserted an optic fiber over the vPAG, where the axon terminals of ChR2-expressing MPA neurons are located (Fig. 5a). The heads of the mice were fitted with a device comprising of a bait object controlled by a servo-motor, a Wi-Fi communication module, and an LED stimulator (Fig. 5b). 24 h prior to day-1, we placed each mouse into the focal fear chamber for 5 min of habituation. After shock was administered on day-1 (post), we performed fear extinction with MIDAS (Fig. 5c). We used MIDAS to guide the mice into the fear-conditioned zone five times during the extinction session by inducing them to chase the head-mounted object through a round-trip pathway, that passed through both the safe zone and the shock zone. As a control, we exposed another group of mice to the chamber without MIDAS for 5 min (Fig. 5d).

To evaluate how MIDAS-induced entries into the fear-conditioned zone affected fear extinction, we compared the effect of exposure between the groups during the extinction test. We found that the MIDAS group showed more robust fear extinction than the control group. The percentage of distance moved, time spent, and number of entries in the shock zone were significantly increased in the MIDAS group after day-6 (Fig. 5e–g). Lower percentage of time spent freezing was noticed in the MIDAS

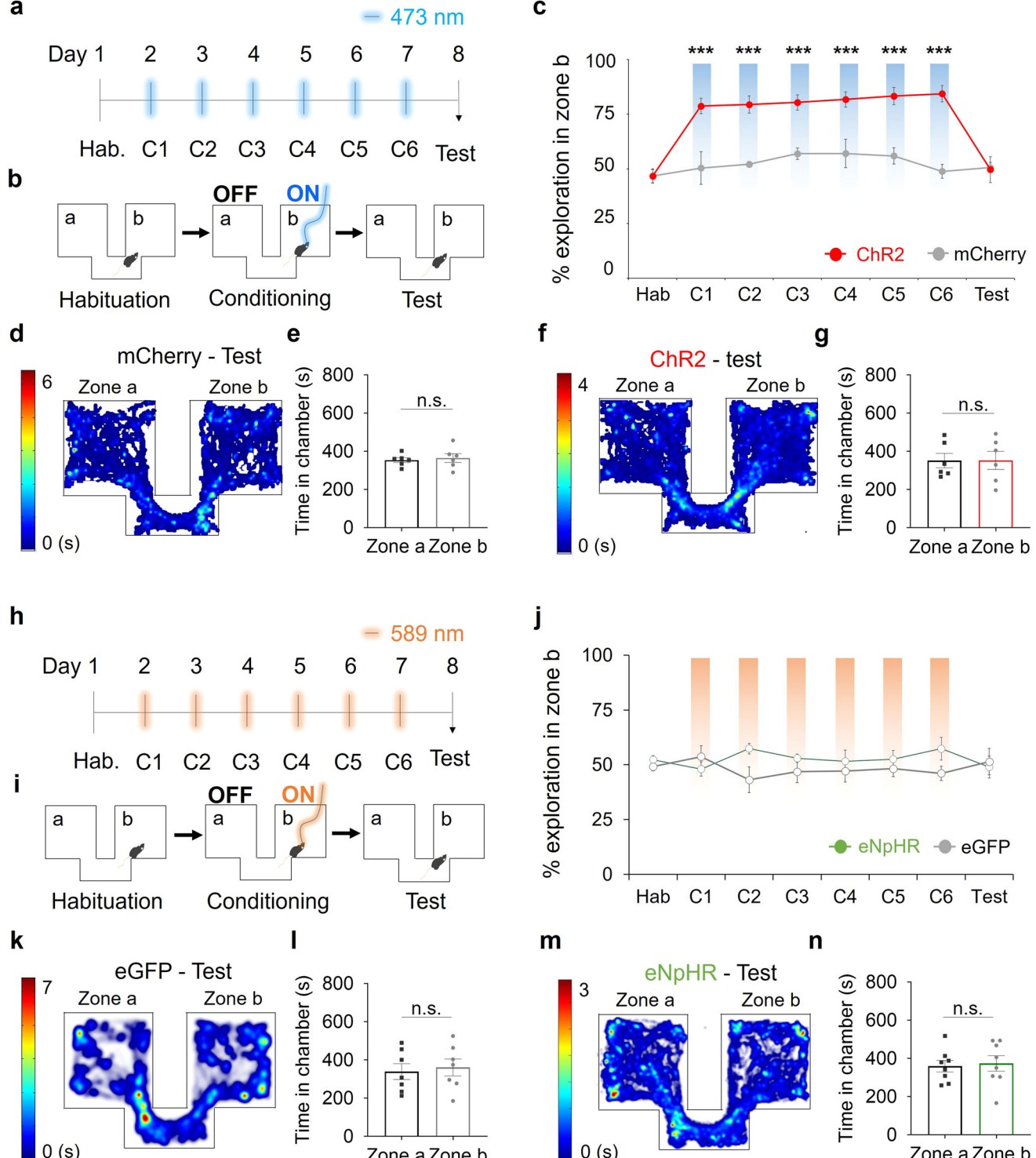

**Fig. 4 MPA-vPAG circuit stimulation increases place exploration behavior. a** The timeline for the CPP test. **b** Custom chamber design and light-stimulation protocol. **c** Exploration of zone b (%) by the ChR2 group (red, $n = 6$) and the mCherry group (gray, $n = 6$; two-way RM ANOVA test, Holm-Sidak post-hoc analysis). **d** A representative heat map of mouse center points locations during the CPP test. **e** Time in zone a vs. zone b for the mCherry group during the CPP test ($n = 6$; paired two-tailed $t$-test). **f** A representative heat map of mouse center points locations during the CPP test. **g** Time in the zone a vs. zone b for the ChR2 group during the CPP test ($n = 6$; paired two-tailed $t$-test). **h** The timeline for the CPP test. **i** Custom chamber design and light stimulation protocol. **j** Exploration of zone b (%) by the eNpHR group (green, $n = 8$) and eGFP group (gray, $n = 7$; two-way RM ANOVA test, Holm-Sidak post-hoc analysis). **k** A representative heat map of mouse center points locations during the CPP test. **l** Time in zone a vs. zone b for the eGFP group during the CPP test ($n = 7$; paired two-tailed $t$-test). **m** A representative heat map of mouse center points locations during the CPP test. **n** Time in zone a vs. zone b for the eNpHR group during the CPP test ($n = 8$; paired two-tailed $t$-test). ***$p < 0.001$. n.s., not significant as $p > 0.05$. All error bars represent s.e.m. Further information on statistical analyses is given in the supplementary data 1.

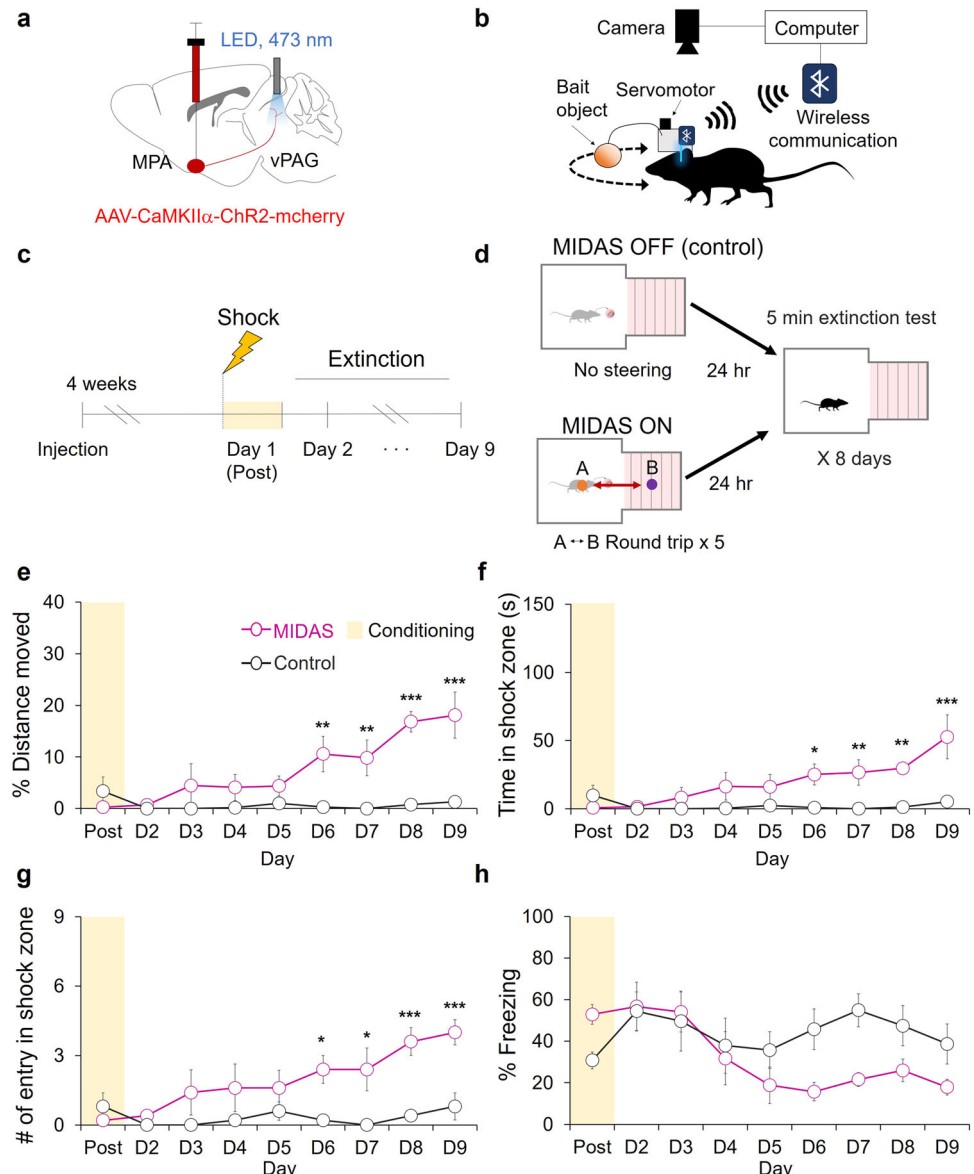

**Fig. 5 MIDAS-driven shock zone exploration induces fear extinction. a** A schematic depiction of virus injection and photostimulation (3 mW, 20 Hz, 5 ms). **b** A schematic depiction of the MIDAS system. **c** The fear-conditioning and extinction test schedules. **d** MIDAS exploration and extinction testing during fear extinction. **e** The percentage of distance moved in the shock zone during the extinction test for the MIDAS group (Magenta, $n = 5$) and the control group (gray, $n = 5$; two-way RM ANOVA test, Holm-Sidak post-hoc analysis). **f** Time in the shock zone during fear extinction for the MIDAS group (Magenta, $n = 5$) and the control group (black, $n = 5$; two-way RM ANOVA test, Holm-Sidak post-hoc analysis). **g** The number of entries into the shock-paired zone during fear extinction for the MIDAS group (Magenta, $n = 5$) and the control group (black, $n = 5$; two-way RM ANOVA test, Holm-Sidak post-hoc analysis). **h** The percentage of freezing into the shock-paired zone during fear extinction for the MIDAS group (Magenta, $n = 5$) and the control group (black, $n = 5$; two-way RM ANOVA test, Holm-Sidak post-hoc analysis). *$p < 0.05$. **$p < 0.01$, ***$p < 0.001$. All error bars represent s.e.m. Further information on statistical analyses is given in the supplementary data 1.

group but not statistically significant (Fig. 5h). These results indicate that object-guided exploration of the fear-conditioned zone facilitates fear extinction.

## Discussion

Fear memories can induce long-term maladaptive fear responses, including avoidance[35–37]. Extinction of these fear memories involves acquiring new memory through exposure to a fear conditioned target without an aversive unconditioned stimulus[1–3]. In fear memory extinction experiments using animal models[7], it has been difficult to verify the effects of exploration because the animals showed avoidance responses to the fear

conditioned target. Thus, the effect of direct exploration on fear memory extinction is not well understood. In previous fear extinction studies, researchers have focused on avoidance and fear responses mediated by the medial prefrontal cortex, amygdala function or hypothalamic stress[38–42], but research on brain circuits that increase the desire to explore have not been conducted for fear extinction.

During PE, patients are exposed to fear-eliciting targets or traumatic memories to extinguish fear. However, the mechanism underlying the curing effects of PE remains unknown[4–6]. To study the mechanism of fear extinction in animal models, many researchers have used a closed cage, which makes it difficult to independently evaluate the effect of approach or escape. Using a

focal fear-conditioning paradigm, we evaluated the approach of a fear-conditioned zone from a non-conditioned zone (Fig. 1) and found that photoactivation of the MPA-vPAG circuit facilitates fear extinction by strengthening approach (Fig. 2, 5).

Studies have shown that photostimulation of reward circuits (e.g., dopaminergic neurons in the ventral tegmental area) increases place preference[43] and modulates fear-related memories[44–47]. Our present results suggest that the role of the MPA-vPAG circuit in fear extinction is unlikely to be due to the modulation of place preference or avoidance (Fig. 4). Instead, the increase in exploration through MPA-vPAG circuit stimulation significantly induced the extinction of the fear memory (Fig. 2e–h and Supplementary Fig. 1). During the MIDAS induced fear extinction test, MIDAS mice experienced photostimulation of the MPA-vPAG circuits equally in both safe and shock zones during round-trip movements between both zones, but showed fear extinction (Fig. 5). Our study suggests that the role of this circuit in inducing fear extinction is due to active exploration, rather than a reward-related effect.

Fear extinction was induced only when photostimulation of the MPA-vPAG circuit increased the explorative activity of mice directly in the shock zone (Figs. 2a–h), but photostimulation of the MPA-vPAG circuit without exploration of the fear-conditioned zone did not facilitate fear extinction (Fig. 2i–k). Direct exploration towards the fear-conditioned zone by photo-stimulation of this circuit, not only increased movement, but also potentially increased the chance to recall the fear memory and facilitate fear extinction. However, photoinhibition of the MPA-vPAG circuit showed similar amounts of exploration and did not affect preference or avoidance during both real-time and conditioning sessions (Supplementary Fig. 3. and Fig. 4h–n). The amount of exploration was not sufficient to affect fear extinction compared to controls (Fig. 3). Consistently, studies have shown that recall of fear memories is required for effective fear extinction, and have implicated the dentate gyrus in memory recall and the attenuation of contextual fear[48,49]. Silencing of fear memory recall was also found to engage dentate gyrus engrams and slow behavioral extinction[50]. Eye Movement Desensitization and Reprocessing (EMDR), a psychotherapeutic method developed by Francine Shapiro[51] to treat symptoms of trauma, also involves recall of fear memory[52–54]. Thus, active exploration towards a fearful target induced by the MPA-vPAG circuit may change the decision-making process depending on the extinction memory through memory recall.

In the present study, we found that MPA-vPAG circuit stimulation facilitated fear extinction through fear-conditioned target exploration and revealed that the fear extinction achieved by prolonged exposure seems to depend heavily on internal states, such as motivation to explore the fear-related target. Animals in the natural environment face a conflict between exploration and avoidance, especially when they need to forage for food. A balance between exploration and fear-circuit mechanisms may contribute to the outcome of this debate. The MPA is a sexually dimorphic region of the brain that is known to have different behaviors depending on sex[22]. However, there were no difference in PE effects according to sex among PTSD patients[55], and in previous studies, exploration or approach function of the MPA did not depend on the sex of rodents[24,33]. Nevertheless, gender differences in MPA mediated fear extinction have to be studied in the future. Our findings suggest that the MPA-vPAG circuit is a pharmacological and psychological target of an efficient method to improve patients' quality of life with emotional disorders.

## Methods

**Animals**. All behavioral experiments were performed on 9- to 11-week-old male C57BL/6 J mice. The mice had free access to food and water and were kept on a 12-

h light-dark cycle at 22 °C. All care and handling of mice was performed according to the directives of the Animal Care and Use Committee of KAIST (Protocol No. KA2020-63). The study received ethical approval (KA2020-65, KA2022-038, KA2022-077-v1).

**Virus injection and fiber-optic cannula implantation**. Mice were anesthetized before surgery with 2,2,2-tribromoethanol via intraperitoneal injection. For activation of the MPA-vPAG circuit, the MPA (AP, +0.2 mm; ML, 0.3 mm; DV, −5.2 mm, from bregma) was injected with 0.5 μl AAV2/9-CaMKIIα-hChR2(H134R)-mCherry (Addgene, USA) or AAV2/5-CaMKIIα-mCherry (University of North Carolina Vector Core, USA). For inhibition, 0.5 μl AAV2/5-CaMKIIα-eNpHR3.0-eYFP (University of North Carolina Vector Core, USA) or AAV2/5-CaMKIIα-eGFP (University of North Carolina Vector Core, USA) was used instead. A fiber-optic cannula (200 μm diameter; Doric Lenses, Canada) was implanted in the vPAG (AP, −4.7 mm; ML, +0.3 mm; DV, −2.6 mm) unilaterally for activation experiments. Dual optic fibers (200 μm diameter, Doric Lenses, Canada) were implanted in the vPAG (AP, −4.7 mm; ML, ±0.3 mm; DV, −2.6 mm) for the inhibition tests. As a last step, the skull was covered with dental cement (Vertex, Netherlands). All mice were given a recovery period of 3–4 weeks before any behavioral experiment was attempted.

**Contextual fear extinction**. All trials were performed in a chamber that was custom made for these experiments. The chamber consisted of two rooms, a safe zone (40 × 40 × 45 cm) and a shock zone (30 × 30 × 45 cm), connected by a hole (30 × 15 cm). The shock zone was equipped with a stainless-steel grid floor connected to a shock generator (Coulbourn Instruments, USA). The non-shock chamber had black and white striped walls intended to help mice distinguish the chambers from one another. Trials were conducted for 10 days. On day-1, each mouse was placed in the center of the safe zone and allowed to explore both zones for 5 min freely. The mouse was trapped in the shock zone, and electrical foot shocks were administered. The foot shock interval was 5 min, and 1 mA foot shock was given 3 times in 15 min. From days-2 through 10, the mice were placed in the center of the safe zone and their activity was recorded for 5 min (extinction test). After the extinction test, the mice of both groups were trapped in the shock zone for 5 min. During this period, the mice were exposed to 20 Hz 5 ms 473 nm light (3 mW) or continuous wavelength 589 nm light (10 mW). For the modified focal zone-fear conditioning test, the mice of both groups were walled in the safe zone for 5 min after the extinction test. The mice received light stimulation (473 nm, 20 Hz, 5 ms) during this period. To test the effect of MIDAS on fear extinction, mice were divided into a MIDAS group (n = 5) and a control group (n = 5) and trials were conducted for 10 days. On day-1, fear conditioning by electrical foot shocks was performed as described above. From days-1 through 9, the mice were placed in the center of the safe zone and their activity was recorded for 5 min. After the extinction test, the behavior of the mice in the MIDAS group was modulated via the MIDAS system. We set the center point of each chamber as a waypoint and guided each mouse on a round trip between the waypoints using the MIDAS monitoring system and navigation algorithm. We repeated this round trip five times per mouse over the course of 5 min. Mice in the control group were exposed to the chamber for the same amount of time without MIDAS application. We automatically analyzed the recording data by EthoVision XT software (Noldus, UK) for the distance moved, time in the shock zone, the number of entries into the shock zone, and freezing (using activity analysis, non-moving state duration of more than 2 s)[56]. We calculated the percentage of distance moved in the shock zone as follows:

$$\text{(Distance moved in shock zone} \times 100)/\text{Total distance moved in both zones.} \quad (1)$$

We also calculated the percentage of time spent freezing, as follows:

$$\text{(Time in freezing} \times 100)/\text{Total recording time.} \quad (2)$$

**Head-mounted control device design**. To remotely induce object-following behavior in mice, we applied the MIDAS system, where a head-mounted device modulates photostimulation and target-object direction using a servomotor[32]. The head-mounted device was comprised of five components: an embedded chipset, an LED module, a target object, a servomotor module, and a battery. The RFD22301 embedded chipset (RFduino, USA) comprised of a microcontroller (ARM Cortex-M3, STM32F101V8T6) and a built-in Bluetooth antenna. The chipset operated the LED and servomotor modules through the MIDAS algorithm, which was programmed in the Arduino IDE (Arduino, 2015). The LED module delivered power to the LED-optic cannula and modulated the frequency of photostimulation over a 0–40 Hz range. In our experiments, we fixed the light intensity at the tip of the optic cannula to 3 mW and used a red Styrofoam sphere (3.5 cm diameter covered with paper) connected to the servomotor via a carbon fiber as the target object. The servomotor (maximum output angle, 180° resolutions, 5°; Hobbyking, USA) steered the movement of the mouse by rotating the target object through its visual field. This device used a 3.7 V lithium-polymer type battery. The entire head-mounted control device (28 × 15 × 21 mm, 23.5 g) was firmly inserted into the external connector of the LED-optic cannula during the experiments and detached afterwards.

**MIDAS algorithm**. Mouse positional information $P(x_m, y_m)$ and the nth waypoint $w_n(x_w, y_w)$ obtained from the CMOS camera (Point Grey, Canada) was analyzed at every time-step (30 ms). In the algorithm, the head angle, $\theta_h$, indicates the angle between a horizontal line and the head direction and the waypoint angle, $\theta_w$, is defined as

$$\theta_w = \tan^{-1}\left(\frac{y_w - y_m}{x_w - x_m}\right), \qquad (3)$$

where $\theta_o$ represents the angle between the object direction and a horizontal line. Depending on the calculated angle, $\theta_w$, the servomotor rotated the target object to the angle closest to $\theta_w$. The target object was made to follow the route to the waypoint, $\theta_o \rightarrow \theta_w$, regardless of mouse movement. Photostimulation (0–40 Hz) via the LED module was activated at

$$|\theta_o - \theta_h| \le 15°. \qquad (4)$$

The MIDAS system algorithm was programmed in Microsoft Visual C++ using the OpenCV library.

**External monitoring system for MIDAS**. A CMOS camera (Point Grey, Canada) above each chamber or maze was used to track mouse paths. The monitoring system computed mouse head positions and angles in real time by applying a color-detection algorithm to the image data acquired via a camera. In addition, a task management tool was used to transmit information to the head-mounted control device via Bluetooth.

**Conditioned place-preference test**. The custom-made place-preference chamber contained two rooms (20 cm × 18 cm) with different patterns on the walls and a corridor between them. EthoVision XT software (Noldus, UK) and a camera (LifeCam, Microsoft, USA) were used to detect and analyze the center point of each mouse to guide the delivery of light stimulation. For CPP tests, we tested the baseline preference on day-1 for 15 min. After allowing the mice to freely explore the chamber, we excluded mice that showed a place preference higher than 65%. During the conditioning session, the mice explored the chamber for 30 min while light stimulation was paired with only one side. The light-paired room was counterbalanced. To avoid excitotoxicity in the activation group, the 473 nm light was turned off when a mouse remained in the light-paired room for longer than 30 s. If the mouse remained in the light-paired room for 1 min after the light was turned off, the light was turned on again. A 589 nm light remained on continuously during the tracking of mouse center points for the inhibition group mice who remained in the light-pair chamber as zone b. On day-8, we recorded 15 min of free exploration without light stimulation. We calculated the percentage of time each mouse spent exploring zone b by dividing the time in the light-paired chamber by the sum of the time in both chambers, as follows:

$$(\text{Time in light} - \text{paired chamber} \times 100)/\text{Time in both chambers}. \qquad (5)$$

**Histology**. Mice were anesthetized and perfused with heparin sodium salt in phosphate-buffered saline (PBS) and then 4% formaldehyde solution. After post-fixation, the brains were sectioned (60 μm thickness) in a vibratome (Leica VT1200S, Leica, Wetzlar, Germany). Brain sections were mounted with Vecta-shield Hardset antifade mounting medium with 4′,6-diamidino-2-phenylindole (DAPI) (Vector Laboratories, Burlingame, CA, United States). Brain sections were imaged under an A1 HD25 high-resolution confocal microscope (Nikon, Tokyo, Japan) and analyzed using NIS-Elements AR analysis software (Nikon, Tokyo, Japan).

**Statistics and Reproducibility**. No statistical analysis was used to predetermine sample sizes. Experimental findings were reliably reproduced among all subjects in all experiments comprised of multiple cohort. Optogenetic experiments were conducted with more than 3 cohorts of animals. All statistical analyses were performed using SigmaPlot (12.0; Systat Software). An unpaired two-tailed *t*-test or Mann-Whitney rank sum test was used to compare data from pairs of experimental groups. Paired two-tailed *t*-tests or Wilcoxon signed-rank tests were used to compare data obtained before and after experiments in the same group. Two-way RM ANOVA tests were performed for multiple comparisons. The test methods and p-values are indicated in the supplementary data 1. All statistical tests were two-sided, and *p*-values < 0.05 were considered statistically significant.

**Reporting summary**. Further information on research design is available in the Nature Portfolio Reporting Summary linked to this article.

## Data availability
Source data and statistical comparisons underlying the main figures are presented in Supplementary Data 1. The raw, unprocessed datasets from the current study were not deposited to a public repository due to their large size but are available from the corresponding author upon request.

## Code availability
The code related to the MIDAS system is available at https://github.com/Daegun-Kim/MIDAS, https://doi.org/10.5281/zenodo.7437597.

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

## Acknowledgements

This work was supported by grants from Samsung Science & Technology (SSTF-BA1301-53). This work was supported by the National Research Foundation of Korea (NRF) grant funded by the Korean government (MSIT) (NRF- 2022R1A2C3013280). This research was supported by the Bio & Medical Technology Development Program of the National Research Foundation (NRF) funded by the Korean government (MSIT) (NRF-2019M3E5D2A01066259).This paper is based on a research which has been conducted as part of the KAIST-funded Global Singularity Research Program for 2022.

## Author contributions

A.S. and S.G.P. designed the experiments; S.G.P., D.G.K., and S.H.B. performed the MIDAS-induced fear extinction test; J.R. and A.S. performed the focal zone-fear paradigm. J.R., A.S., K.S, and J.L. performed the conditioned place preference test; J.R., A.S., S.G.P., and D.G.K. analyzed the data; A.S. and J.R. drafted the manuscript; D.K. edited the manuscript; and all authors reviewed and approved the manuscript.

## Competing interests

The authors declare no competing interests.
