## [Peer Review File · Communications Biology]

Reviewers' comments:

Reviewer #1 (Remarks to the Author):

In their present manuscript „Exploration driven by a medial preoptic circuit facilitates fear extinction“ Shin and colleagues analyzed to role of an MPA to vPAG circuit during context conditioning and extinction learning in a two-compartment chamber design. The authors can demonstrate that these effects are linked to increased explorative behavior and not by place preference induced by photostimulation of MPA terminals in the vPAG. Overall, the work is very interesting and covers an important topic. The methods used for circuit analysis/manipulation are state-of-the-art. The MIDAS guided behavior combined with optogenetic stimulation is a very interesting, clever designed approach.

Although the study is well designed and the findings are very interesting, I would like to raise a couple of issues:

1. The MPA has been shown to be a highly sexually dimorphic with substantial differences between male and females (rodents and humans alike). Why have only male mice been used in this study? Given the high prevalence for anxiety disorders in (human) females, this study, which aims to understand/improve exposure therapy, should have female mice included. Can the authors at least provide a good reason for this restriction?
2. The data sets in e.g. Fig. 2 e, g, h and i have been tested using a two-way ANOVA. To my understanding a repeated measurement ANOVA would be appropriate in this case, testing several consecutive observations.
3. The experiments for Fig. 2 were done using a “no-light” control group. Later on, the control group was injected with an mCherry vector only and the mice received light. Why is there this switch in controls? In my opinion, the mCherry-only + light would have been the better control group throughout the experiments. If mice are able to see light stimuli through the optic fiber (e.g. by reflections or by looking up), this would be an additional cue in the first set of experiments (Fig.2) which would be missing in the “no light” control group.
4. The conclusion of the last experiment is that “..object-guided exploration of the fear conditioned place facilitates fear extinction”. I have some problems to follow this conclusion since the “control group” seems not to enter the shock context to an amount, which would allow extinction in the shock context in absence of the US. The MIDAS data are compared to a group, which does not extinguish at all.
5. During establishment of the focal place-fear paradigm freezing was analyzed. In the following experiments freezing data are not shown. Is there a reason for omitting freezing data from analysis?

Reviewer #2 (Remarks to the Author):

In this manuscript, Shin and colleagues investigate the role of activating medial preoptic area (MPA) to ventral periaqueductal gray (vPAG) projection in extinction of fear avoidance. To this end, the authors performed a fear avoidance task which involved mice undergoing contextual fear conditioning and investigated the role of activating CaMKIIa-positive MPA neuron projections to vPAG in extinction of fear avoidance behavior. The authors find that optogenetic activation of the MPA-vPAG projection in the fear conditioned chamber following contextual conditioning resulted in facilitated extinction of avoidance behavior. On the other hand, activation or inhibition of the MPA-vPAG projection did not cause place preference or avoidance, respectively. Finally, the authors performed a MPA-vPAG circuit photostimulation-driven chasing of a head-mounted object and found that directing animals to make round trips between the neutral and the fear conditioned chambers enhanced extinction. Based on these results the authors conclude that exploration driven by activation of the MPA-vPAG circuit facilitates fear extinction. Understanding the neural circuits underlying extinction of fear avoidance behavior is an important question with clinical significance. However, there are a number of major

concerns relating to the experimental design as well as interpretation of the results that needs to be addressed.

Major concerns:

1) The main experiments (Figure 2 and Figure 4) lack mCherry control groups. Previous studies have shown that laser light can cause nonspecific effects by inducing tissue heating (for example see Stujenske et al., 2015 Cell Reports). Therefore, it is possible that the observed effects of laser stimulation might be due to a nonspecific effect of the laser. Considering laser sessions are quite long (5 min in Figure 2), it is necessary to have mCherry control groups to rule this possibility out. If the authors have run experiments in Figure 2 and 4 in the absence of mCherry control groups, they need to repeat these experiments in the presence of mCherry groups. Ideally, the animals comprising the mCherry and ChR2 groups should be from same batches (i.e. littermates) that undergo all the procedures (surgeries, behavior tests, optogenetic stimulation, etc) simultaneously to control for all possible nonspecific factors.

2) The authors are optogenetically activating the CamKIIa expressing MPA neuron terminals in vPAG and induce increased mobility/exploration which leads to enhanced extinction of fear avoidance behavior. Activation of these glutamatergic MPA terminals likely leads to increased firing in vPAG neurons. However, does activation of specifically the MPA projection necessary or whether enhancing any glutamatergic input to vPAG would be sufficient to affect fear extinction? Importantly, how the activity of CamKII positive MPA neurons changes during fear conditioning and extinction and whether CamKII expressing MPA neurons are indeed involved in fear extinction is not clear. That artificially activating a glutamatergic input to vPAG is sufficient to affect extinction of fear avoidance does not necessarily mean that this circuit is required for fear extinction under normal circumstances. It is important to test whether MPA to vPAG circuit is necessary for fear extinction. It would be good if authors perform optogenetic inhibition of MPA to vPAG projection in a setting similar to the ChR2 experiment in Figure 2 and examine the effects of inhibiting this circuit on fear extinction. It would be important to see these results because in Figure 3, although optogenetic excitation is sufficient to cause enhanced mobility/exploration (Fig 3b), optogenetic inhibition does not seem to have an overall significant effect on exploration (Fig 3g) suggesting that although artificially enhancing activity in this circuit by opto stimulation is sufficient to enhance mobility/exploration, opto-inhibition of the same circuit has no significant effect. If this circuit was important for exploration, one would have expected to see a significant effect of inhibiting this circuit.

3) In Figure 2, laser stimulation is performed in the fear conditioned chamber leading to enhanced mobility in this chamber. One important question is whether activation of the MPA to vPAG circuit in specifically the conditioned chamber is a necessary condition to observe facilitated extinction of fear avoidance behavior. For instance, can any nonspecific increased mobility result in similar effects on extinction? It is expected that exploration of specifically the conditioned chamber, but not any increased mobility, should lead to enhanced extinction. However, a control group testing this issue is missing in the study. To address this question, the authors can perform the same experiment but this time by activating the MPA to vPAG circuit in the safe chamber rather than the conditioned chamber and see whether this manipulation would have an effect on extinction. And, if stimulation and enhanced mobility leads in the safe chamber leads to extinction of fear avoidance, what would these results suggest?

4) Similar to point 2 above, in Figure 4, the authors should add a control group where they steer the animals using MIDAS but confine them to the safe chamber rather than making round trips between the safe and conditioned chamber and see whether entering the conditioned chamber is a necessary condition for the observed effects on extinction.

5) The authors are performing a fear avoidance paradigm however the literature on fear avoidance (i.e. inhibitory avoidance, active avoidance, etc) and the neural circuits involved in fear avoidance and

extinction of avoidance behavior are not cited adequately in the manuscript. This literature needs to be included.

6) Histology pictures in Figures 2b and 3e are not very clear. A larger section of the coronal slice needs to be shown to see the whole extend of virus expression and see how far the virus spread in MPA. Furthermore, the authors should include histological verification of optic fiber placements and virus expression profiles of all mice in each of the experiments as supplementary figures.

Reviewer #3 (Remarks to the Author):

1. General comments and recommendation.

This manuscript shows interesting concept, fear extinction by modulating medial preoptic excitatory neurons, and activating the neurons at PAG which plays a role in motivation and pain modulation. However, it would not be clear that MPA circuit would be related with fear extinction. MPA has been studied that this area would regulate social behavior. If fear extinction is facilitated with suppressing PAG and activating MPA, it can tell that MPA has a role in fear extinction. Also, MPA-PAG synapses are not determined which synapses are critical to fear extinction, excitatory-excitatory or excitatory-inhibitory neurons. If the synapses are found, it could be possible to find out treatment for PTSD patients.

2. Major comments

- 1) More background about MPA and PAG brain areas in introduction part would be better let us to understand your study.
- 2) Whenever mice got the photostimulation in the zone b, mice spent more time in the zone b in figure 3. The MPA circuit may not connect memory circuit, but it does not mean that MPA is not related with reward or preference.
- 3) For the fear extinction, fear-conditioned place and object-guided exploration were done. Could fear extinction happen with stimulating MPA-vPAG in non-fear conditioned place?
- 4) In your study, there was no 'control virus (such as AAV-CamKii α -DIO-mcherry)' control group. For getting rid of effect of laser stimulation, using control virus would be helpful.
- 5) In ChR2 result, it seems like that activating PAG causes more movement as well as fear extinction. It needs more explanation.
- 6) Figure 2 (c) needs more detail explanation. Could you describe more detail experimental paradigm?
- 7) In MIDAS, is head-direction related with fear-condition?

3. Minor comments (list them, toward specific contents)

- 1) Distance moved (m) chart could be changed to distance moved ratio (fear box/ total or non-fear box).
- 2) NpHR carrying virus were injected to both hemispheres MPA, which is different from ChR2 injected to one hemisphere MPA.
- 3) In figure 2 (c), the figure has no "yellow indication, conditioning" and pre-test.

Reviewer#1

In their present manuscript "Exploration driven by a medial preoptic circuit facilitates fear extinction" Shin and colleagues analyzed the role of an MPA to vPAG circuit during context conditioning and extinction learning in a two-compartment chamber design. The authors can demonstrate that these effects are linked to increased explorative behavior and not by place preference induced by photostimulation of MPA terminals in the vPAG. Overall, the work is very interesting and covers an important topic. The methods used for circuit analysis/manipulation are state-of-the-art. The MIDAS guided behavior combined with optogenetic stimulation is a very interesting, clever designed approach.

Although the study is well designed and the findings are very interesting, I would like to raise a couple of issues:

1) The MPA has been shown to be a highly sexually dimorphic with substantial differences between male and females (rodents and humans alike). Why have only male mice been used in this study? Given the high prevalence for anxiety disorders in (human) females, this study, which aims to understand/improve exposure therapy, should have female mice included. Can the authors at least provide a good reason for this restriction?

Thanks for the nice comment. We also think it is crucial to study female cases. But previous studies showed that there were no differences in the effects of PE according to sex among PTSD patients (ref 55 of manuscript), and the exploration or approach function of the MPA was not dependent on the sex of rodents (ref 24 and 33 of manuscript). Nevertheless, although it is crucial to study female cases, we mentioned the necessity of research in the discussion section rather than direct experimentation due to the limited time and resources. If our conclusion is accepted, we would like to study this great question on sex differences more deeply as an independent study.

2) The data sets in e.g. Fig. 2 e, g, h and i have been tested using a two-way ANOVA. To my understanding a repeated measurement ANOVA would be appropriate in this case, testing several consecutive observations.

Thanks for the comment. We reanalyzed the data in Figure 2e–h using two-way repeated measures ANOVA and obtained the same results. Photoactivation of the MPA^{CaMKIIa}-vPAG circuit significantly induced fear extinction by exploration.

3) The experiments for Fig. 2 were done using a "no-light" control group. Later on, the control group was injected with an mCherry vector only and the mice received light. Why is there this switch in controls? In my opinion, the mCherry-only + light would have been the better control group throughout the experiments. If mice are able to see light stimuli through the optic fiber (e.g. by reflections or by looking up), this would be an additional cue in the first set of experiments (Fig.2) which would be missing in the "no light" control group.

Thanks for the nice comment. We agree with your comments and have conducted additional experiments. We injected AAV-CaMKIIa-mCherry virus in the MPA and implanted optic fibers into the PAG and then performed the same experiment in Figure 2 to exclude the effect of light. We got the same results as previously. As analyzed using a two-way RM ANOVA, photoactivation of the MPA^{CaMKIIa}-vPAG group significantly induced fear extinction compared to the mCherry virus injected

control group.

4) The conclusion of the last experiment is that “..object-guided exploration of the fear conditioned place facilitates fear extinction”. I have some problems to follow this conclusion since the “control group” seems not to enter the shock context to an amount, which would allow extinction in the shock context in absence of the US. The MIDAS data are compared to a group, which does not extinguish at all.

Thanks for the nice comment. As seen from control group results, mice in their natural state rarely enter a place with fear. Since we wanted to know that entering a fear-conditioned zone causes fear extinction, we compared the control group with little place exposure due to their fear of the place. The group were exposed to the fear zone via MIDAS. We were able to reveal that entry (exploration) of the fear-conditioned zone facilitates the extinction of fear, as shown in Figure 5.

5) During establishment of the focal place-fear paradigm freezing was analyzed. In the following experiments freezing data are not shown. Is there a reason for omitting freezing data from analysis?

Thanks for the meaningful comment. We used EthoVision program to analyze freezing data automatically and added new figures (Figure 1h, Figure 2h, Figure 2k, Figure 3g, and Figure 5h). As a result, we confirmed that photoactivation of the MPA-vPAG circuit in the fear-conditioned zone reduced the freezing rate.

Reviewer #2 (Remarks to the Author):

In this manuscript, Shin and colleagues investigate the role of activating medial preoptic area (MPA) to ventral periaqueductal gray (vPAG) projection in extinction of fear avoidance. To this end, the authors performed a fear avoidance task which involved mice undergoing contextual fear conditioning and investigated the role of activating CaMKIIa-positive MPA neuron projections to vPAG in extinction of fear avoidance behavior. The authors find that optogenetic activation of the MPA-vPAG projection in the fear conditioned chamber following contextual conditioning resulted in facilitated extinction of avoidance behavior. On the other hand, activation or inhibition of the MPA-vPAG projection did not cause place preference or avoidance, respectively. Finally, the authors performed a MPA-vPAG circuit photostimulation-driven chasing of a head-mounted object and found that directing animals to make round trips between the neutral and the fear conditioned chambers enhanced extinction.

Based on these results the authors conclude that exploration driven by activation of the MPA-vPAG circuit facilitates fear extinction. Understanding the neural circuits underlying extinction of fear avoidance behavior is an important question with clinical significance. However, there are a number of major concerns relating to the experimental design as well as interpretation of the results that needs to be addressed.

Major concerns:

1) The main experiments (Figure 2 and Figure 4) lack mCherry control groups. Previous studies have shown that laser light can cause nonspecific effects by inducing tissue heating (for example see Stujenske et al., 2015 Cell Reports). Therefore, it is possible that the observed effects of laser stimulation might be due to a nonspecific effect of the laser. Considering laser sessions are quite long (5 min in Figure 2), it is necessary to have mCherry control groups to rule this possibility out. If the authors have run experiments in Figure 2 and 4 in the absence of mCherry control groups, they need to repeat these experiments in the presence of mCherry groups. Ideally, the animals comprising the mCherry and ChR2 groups should be from same batches (i.e. littermates) that undergo all the

procedures (surgeries, behavior tests, optogenetic stimulation, etc) simultaneously to control for all possible nonspecific factors.

Thanks for the nice comment. We agree with your comments and have conducted additional experiments. We injected AAV-CaMKIIa-mCherry virus in the MPA and implanted optic fibers into the PAG and then performed the same experiment in Figure 2 to exclude the effect of light. We got the same results as previously. As analyzed using a two-way RM ANOVA, photoactivation of the MPA^{CaMKIIa}-vPAG group significantly induced fear extinction compared to the mCherry virus injected control group. In this Figure 2, we observed no laser effects on the experiment. Therefore, we also think that the MIDAS experiment in Figure 4 also does not have any non-specific light effects in the same experiment task.

2) The authors are optogenetically activating the CamKIIa expressing MPA neuron terminals in vPAG and induce increased mobility/exploration which leads to enhanced extinction of fear avoidance behavior. Activation of these glutamatergic MPA terminals likely leads to increased firing in vPAG neurons. However, does activation of specifically the MPA projection necessary or whether enhancing any glutamatergic input to vPAG would be sufficient to affect fear extinction? Importantly, how the activity of CamKII positive MPA neurons changes during fear conditioning and extinction and whether CamKII expressing MPA neurons are indeed involved in fear extinction is not clear. That artificially activating a glutamatergic input to vPAG is sufficient to affect extinction of fear avoidance does not necessarily mean that this circuit is required for fear extinction under normal circumstances. It is important to test whether MPA to vPAG circuit is necessary for fear extinction. It would be good if authors perform optogenetic inhibition of MPA to vPAG projection in a setting similar to the Chr2 experiment in Figure 2 and examine the effects of inhibiting this circuit on fear extinction. It would be important to see these results because in Figure 3, although optogenetic excitation is sufficient to cause enhanced mobility/exploration (Fig 3b), optogenetic inhibition does not seem to have an overall significant effect on exploration (Fig 3g) suggesting that although artificially enhancing activity in this circuit by opto stimulation is sufficient to enhance mobility/exploration, opto-inhibition of the same circuit has no significant effect. If this circuit was important for exploration, one would have expected to see a significant effect of inhibiting this circuit.

Thanks for the meaningful comment. We performed optogenetic inhibition of the MPA-vPAG circuit in a similar setting to the Chr2 experiment in Figure 2a-2h. In Supplementary Figure 3, eNpHR mice did not significantly differ in exploration sessions with control mice. Also, fear extinction by exploration did not significantly change in the photoinhibition test in figure 3.

3) In Figure 2, laser stimulation is performed in the fear conditioned chamber leading to enhanced mobility in this chamber. One important question is whether activation of the MPA to vPAG circuit in specifically the conditioned chamber is a necessary condition to observe facilitated extinction of fear avoidance behavior. For instance, can any nonspecific increased mobility result in similar effects on extinction? It is expected that exploration of specifically the conditioned chamber, but not any increased mobility, should lead to enhanced extinction. However, a control group testing this issue is missing in the study. To address this question, the authors can perform the same experiment but this time by activating the MPA to vPAG circuit in the safe chamber rather than the conditioned chamber and see whether this manipulation would have an effect on extinction. And, if stimulation and enhanced mobility leads in the safe chamber leads to extinction of fear avoidance, what would these results suggest?

Thanks for the nice comment. If fear extinction is induced by exploration in the fear-conditioned

place, it would be highly effective in treating actual patients. So, we conducted photoactivation of the MPA-vPAG circuits in a safe zone. Photoactivation of this circuit increased the mobility and exploration in the safe zone (Supplementary Figure 2a-b). However, there were no significant differences with the control group in the fear-extinction test (Figure 2i-2k and Supplementary Figure 2c-2e). For these results, the fear extinction effects by exploration are induced only by exploring for the fear-conditioned zone, not increase in mobility,.

4) Similar to point 2 above, in Figure 4, the authors should add a control group where they steer the animals using MIDAS but confine them to the safe chamber rather than making round trips between the safe and conditioned chamber and see whether entering the conditioned chamber is a necessary condition for the observed effects on extinction.

That is quite an interesting point. We genuinely want to perform that experiment. But the purpose of the MIDAS experiment was to increase the number of entries into the fear-conditioned zone. In the focal zone-fear paradigm, we performed photostimulation of the MPA^{CaMKIIa}-vPAG circuit while locking them in the safe chamber. There was no significant difference between ChR2 and mCherry groups. So, we thought that the MIDAS experiment would confine mice to the safe zone to show the same results as the focal zone-fear paradigm, with photoactivation in the safe chamber.

5) The authors are performing a fear avoidance paradigm however the literature on fear avoidance (i.e. inhibitory avoidance, active avoidance, etc) and the neural circuits involved in fear avoidance and extinction of avoidance behavior are not cited adequately in the manuscript. This literature needs to be included.

Thanks for the nice comment. This comment made the discussion part easier to understand. We added the following sentences and references to the discussion.

"Fear memories can induce long-term maladaptive fear responses, including avoidance³⁵⁻³⁷. Extinction of these fear memories involves acquiring new memory through exposure to a fear conditioned target without an aversive unconditioned stimulus^{1, 2, 3}. In fear memory extinction experiments using animal models⁷, it has been difficult to verify the effects of exploration because the animals showed avoidance responses to the fear conditioned target. Thus, the effect of direct exploration on fear memory extinction is not well understood. In previous fear extinction studies, researchers have focused on avoidance and fear responses mediated by the medial prefrontal cortex (mPFC), amygdala function or hypothalamic stress³⁸⁻⁴², but research on brain circuits that increase the desire to explore have not been conducted for fear extinction."

And also added the following sentences and references to the introduction part,

"Also, the ventral periaqueductal gray (vPAG), which receives a strong input from the MPA^{26, 27}, is well known to be involved in modulating approaching (fight) or avoiding (flight) behaviors^{28,29} and hunting behaviors^{30, 31} "

6) Histology pictures in Figures 2b and 3e are not very clear. A larger section of the coronal slice needs to be shown to see the whole extend of virus expression and see how far the virus spread in MPA. Furthermore, the authors should include histological verification of optic fiber placements and virus expression profiles of all mice in each of the experiments as supplementary figures

Thanks for the comment. We changed the histology pictures for ChR2 (Fig. 2b) and eNpHR groups (Fig. 3c). And in the supplementary figure, we added histology pictures for control groups, mCherry (Supplementary Fig. 1a) and eGFP groups (Supplementary Fig. 3a). We are able to show histology for control experiments, current ChR2 experiments and eNpHR experiments. Unfortunately, the ChR2 experiments were done more than 6 years ago and the histological have been discarded already. Our laboratory procedure always involves checking and excluding mice that have optic fibers that are misplaced or viral expression that spreads outside of the region of interest. Also, in the Park *et al*/study, even if the virus spread around the MPA, it will be okay because there is very little projection to the PAG from sites other than the MPA.

- Show in the supplementary Figure 4 in the Park, Sae-Geun, et al. "Medial preoptic circuit induces hunting-like actions to target objects and prey." *Nature neuroscience* 21.3 (2018): 364-372".

Reviewer #3 (Remarks to the Author):

General comments and recommendation.

This manuscript shows interesting concept, fear extinction by modulating medial preoptic excitatory neurons, and activating the neurons at PAG which plays a role in motivation and pain modulation. However, it would not be clear that MPA circuit would be related with fear extinction. MPA has been studied that this area would regulate social behavior. If fear extinction is facilitated with suppressing PAG and activating MPA, it can tell that MPA has a role in fear extinction.

Also, MPA-PAG synapses are not determined which synapses are critical to fear extinction, excitatory-excitatory or excitatory-inhibitory neurons. If the synapses are found, it could be possible to find out treatment for PTSD patients.

Major comments

1) More background about MPA and PAG brain areas in introduction part would be better let us to understand your study.

Thanks for the nice comment. The comment made the introduction easier to understand. We added the following sentences and references to the introduction.

"The medial preoptic area (MPA) is known to regulate approach behaviors and exploratory interactions with a variety of targets²¹⁻²⁵. Also, the ventral periaqueductal gray (vPAG), which receives

a strong input from the MPA^{26, 27}, is well known to be involved in modulating approaching (fight) or avoiding (flight) behaviors^{28,29} and hunting behaviors^{30, 31}. Recent studies have revealed that MPA projections to the vPAG mediate novelty seeking, exploration, hunting behaviors, and approach to a variety of targets³²⁻³⁴."

2) Whenever mice got the photostimulation in the zone b, mice spent more time in the zone b in figure 3. The MPA circuit may not connect memory circuit, but it does not mean that MPA is not related with reward or preference.

Thanks for the comment. In figure 4, we tested whether fear extinction induced by MPA circuit stimulation was caused by exploration of the fear place or a rewarding effect of the circuit using a conditioned place preference test. As described in previous reports, when a region related to a reward circuit, such as dopaminergic neurons in the ventral tegmental area are photoactivated, preference for a specific place occurs in the post-test without light stimulation, after conditioning. Our results show that conditioned preference did not appear in the post-test, meaning that stimulation of the MPA circuit does not induce preference conditioning for a specific place and that the fear extinction effect is not due to an increase in preference.

- Ref) Tsai, H.-C. et al. Phasic firing in dopaminergic neurons is sufficient for behavioral conditioning. *Science* 324, 1080-1084 (2009)

3) For the fear extinction, fear-conditioned place and object-guided exploration were done. Could fear extinction happen with stimulating MPA-vPAG in non-fear conditioned place?

Thanks for the nice comment. If fear extinction is induced by exploration in the fear-conditioned place, it would be highly effective in treating actual patients. So, we conducted photoactivation of the MPA-vPAG circuits in a safe zone. Photoactivation of this circuit increased the mobility and exploration in the safe zone (Supplementary Figure 2a-b). However, there were no significant differences with the control group in the fear-extinction test (Figure 2i-2k and Supplementary Figure 2c-2e). For these results, the fear extinction effects by exploration are induced only by exploring for the fear-conditioned zone, not increase in mobility.

4) In your study, there was no 'control virus (such as AAV-CamKIIa-DIO-mcherry)' control group. For getting rid of effect of laser stimulation, using control virus would be helpful.

Thanks for the nice comment. We agree with your comments and have conducted additional experiments. We injected AAV-CaMKIIa-mCherry virus in the MPA and implanted optic fibers into the PAG and then performed the same experiment in Figure 2 to exclude the effect of light. We got the same results as previously. As analyzed using a two-way RM ANOVA, photoactivation of the MPA^{CaMKIIa}-vPAG group significantly induced fear extinction compared to the mCherry virus injected

control group.

5) In Chr2 result, it seems like that activating PAG causes more movement as well as fear extinction. It needs more explanation.

Thanks for the nice comment. Photoactivation of this circuit increased the mobility and exploration in the safe zone (Supplementary Figure 2a-b). However, there were no significant differences with the control group in the fear-extinction test (Figure 2i-2k and Supplementary Figure 2c-2e). For these results, the fear extinction effects by exploration are induced only by exploring for the fear-conditioned zone, not increase in mobility.

6) Figure 2 (c) needs more detail explanation. Could you describe more detail experimental paradigm?

We modified the description of figure 2c in the article by changing the control group to the mCherry group. We added a more detailed explanation about the focal place-fear paradigm. In the article, we changed the sentence to "To conduct the focal zone-fear paradigm task, mice underwent daily exploration sessions from day-2 (post) to day-9: mice received light stimulation in the shock zone for 5 minutes. Twenty-four hours after each exploration session, mice from both groups were subjected to an exploration test in which they were allowed to roam freely throughout the focal fear chamber for 5 minutes as the extinction session (Fig. 2c)".

7) In MIDAS, is head-direction related with fear-condition?

In the MIDAS experiment, the direction of the head of the MIDAS ON group was programmed to face a predetermined way as described in the methods section. As a result, the orientation of the head was always automatically adjusted on all MIDAS ON mice. Head-direction of mice was related to object positioning, the only purpose of which was to induce exploration of the fear zone and thus in turn facilitate fear extinction.

Minor comments (list them, toward specific contents)

1) Distance moved (m) chart could be changed to distance moved ratio (fear box/ total or non-fear box).

Thanks for the nice suggestion. We modified the distance moved (m) as the percentage of distance moved, which means $(\text{Distance moved in shock place} \times 100) / \text{Total distance moved in both zones}$. We applied the suggested method in Figures 1g, 2e, 2j, 3d, and 5d.

2) NpHR carrying virus were injected to both hemispheres MPA, which is different from Chr2 injected to one hemisphere MPA.

We thank the reviewer for asking this question. In general, unilateral activation of brain nuclei is sufficient to produce a phenotype whereas during inhibition, normally there is a redundancy between the hemispheres, which means that both hemispheres must be inactivated in order to produce proper inhibition of a brain function:

References

Park, Sae-Geun, et al. "Medial preoptic circuit induces hunting-like actions to target objects and prey." *Nature neuroscience* 21.3, 364-372 (2018).

Tan, Na, et al. "Lateral Hypothalamus Calcium/Calmodulin-Dependent Protein Kinase II α Neurons Encode Novelty-Seeking Signals to Promote Predatory Eating." *Research* 2022 (2022).

3) In figure 2 (c), the figure has no "yellow indication, conditioning" and pre-test

We modified figure 2c, including a yellow indication as conditioning and pre-test.

REVIEWERS' COMMENTS:

Reviewer #1 (Remarks to the Author):

I would like to thank the authors for addressing all my concerns carefully.
I have no further concerns. I recommend this manuscript for publication.

Reviewer #2 (Remarks to the Author):

The authors have responded to my comments adequately. I have no further issues.

Reviewer #3 (Remarks to the Author):

The authors have satisfactorily addressed most of my concerns. In particular, the authors have added control experiment and explained why their result is related with fear-extinction, not reward. The only remaining concerns are minor.

1. In Abstract, 27th line "...a fearful target and suggests..." should be "suggest".
2. In introduction, it is not smoothly connected those two 40th line and 41st line.
3. In result Fig. 1a, is there any reason to make those two rooms with different size?
4. Was the foot shock interval 2.5 min?
5. 68th line (Fig.1c) not (Fig 1d).
6. 68th line, is it not important how long they took for the first entry?
7. 83rd line, isn't it better to switch Fig.2c and Fig. 1b?
8. 85th line, was supplementary experiment conducted with the same mice as those for Fig 2 d? If it was, Has fig 2d task done before the supplementary figure

Reviewer #1 (Remarks to the Author):

I would like to thank the authors for addressing all my concerns carefully. I have no further concerns. I recommend this manuscript for publication.

Reviewer #2 (Remarks to the Author):

The authors have responded to my comments adequately. I have no further issues.

Reviewer #3 (Remarks to the Author):

The authors have satisfactorily addressed most of my concerns. In particular, the authors have added control experiment and explained why their result is related with fear-extinction, not reward. The only remaining concerns are minor.

1. In Abstract, 27th line "...a fearful target and suggests..." should be "suggest".

Thanks. We modified the sentence.

2. In introduction, it is not smoothly connected those two 40th line and 41st line.

For smooth connection, we modified the sentence.

From "Unveiling the neural mechanism that induces motivation to explore a fear-eliciting target may facilitate the development of additional efficient exposure therapies and/or drugs."

to "Unraveling the neural mechanism that induces motivation to explore a fear-eliciting target. Which can reduce this avoidance, may facilitate the development of additional efficient exposure therapies and/or drugs.".

3. In result Fig. 1a, is there any reason to make those two rooms with different size?

To increase the entrance to the shock zone. Mice prefer the corner.

4. Was the foot shock interval 2.5 min?

The foot shock interval is 5 min. We give 1mA foot shock 3 times in 15 minutes.

5. 68th line (Fig.1c) not (Fig 1d).

For Fig.1c, we show the scheme of the foot shock. In Fig.1d, we want to show the difference in mice behavior for 5 minutes before(pre) and after(post) the foot shock.

6. 68th line, is it not important how long they took for the first entry?

The latency of the first entry is also important. However, we thought that fear extinction could be demonstrated even with the four factors: Time in shock zone(s), number of entries, percentage of distance moved inside the shock zone, and percentage of freezing.

7. 83rd line, isn't it better to switch Fig.2c and Fig. 1b?

The top of Fig.2c shows the same timeline as Fig.1b. We want to highlight the light-given or not and the zone that mice explore during exploration or extinction sessions.

8. 85th line, was supplementary experiment conducted with the same mice as those for Fig 2 d? If it was, Has fig 2d task done before the supplementary figure

The same mice were used in both Fig.2a-2h and Supplementary Fig. 1a-1c. After the foot shock was given, the exploration and extinction tests were repeated for 8 days. Fig.2d task has been done parallel.